# How sampling affects training: an effective sampling theory study for long-tailed image classification

## Abstract

The long-tailed image classification problem has been very challenging for a long time. Suffered from the unbalanced distribution of categories, many deep vision classification methods perform well in the head classes while poor in the tail ones. This paper proposes an effective sampling theory, attempting to provide a theoretical explanation for the decoupling representation and classifier for long-tailed image classification. To apply the above sampling theory in practice, a general jitter sampling strategy is proposed. Experiments show that variety of long-tailed distribution algorithms exhibit better performance based on the effective sampling theory. The code will be released soon later.

## 1 Introduction

The image classification problems are fundamental tasks in computer vision, and many methods based on deep learning have achieved gratifying results on artificially constructed datasets so far. However, due to the large discrepancy between distributions for different classes, the classification model performs very well for head categories, but usually gives an inaccurate prediction for the tail ones at the same time. This phenomena dose not only occurs in image classification, but also in other common vision tasks such as semantic segmentation He et al. (2021); Wang et al. (2020a), object detection Ouyang et al. (2016); Li et al. (2020) and so on.

Researches on long-tail classification problems mainly focus on the following research perspectives including loss function re-weighting Cao et al. (2019), training data re-sampling Mahajan et al. (2018), and transfer learning strategies in embedding level Liu et al. (2020). The main idea solving the imbalanced classification problem is to enhance the training proportion for the tail categories so as to alleviate the overfitting for the head ones. Kang et al. (2019) points out the strong dependence between the representation learning for backbone network and classifier learning for the the last fully connected layer, and concludes that the optimal gradient for training the backbone network and classifier are obtained from the original sampling distribution and re-sampling distribution such as class-balanced sampling respectively, from which the mainstream of two-stage optimization strategy is gradually accepted by more researchers. Xiang et al. (2020) further alleviates the strong dependence of the single-expert model with a specific training distribution, leading to an improvement of classification accuracy both for head and tail categories.

Kang et al. (2019); Zhou et al. (2020) mentions that the mainstream methods for long-tailed distribution requires two stages learning. Sampling process need be conducted within the original distribution to learn in the first step stage for representation, without an ample theoretical explanation for this phenomena however. Inspired by Cui et al. (2019), we realised that the growth between the actual effective samples and the actual number of samples does not change synchronously in the first training stage, where the effective sample growth formula is given by Cui et al. (2019). Based on the concept of effective sample, our expanded effective sampling theory is proposed. Here we give two important findings. **The total number of effective samples** is the primary factor affecting the training for long-tailed distribution, and the second one is the **effective sample utilization**.The improvement of accuracy on the long-tailed distribution can be achieved through the process of maximizing the total number of effective samples and balancing the effective samples utilization among categories.

The main **contributions** of this paper are as follows:

1. We build a complete theory on effective sampling, which could be used for studying the properties of sampling with/without replacement, through which the optimal sampling methods are proposed.

2. A general jitter sampling strategy is proposed for the piratical application, and experiments on various public datasets have been carried out. The experimental results reach the competitive performance which further verify the core factor of our theory, that is, the **total number of effective samples** is the core factor affecting the first learning stage and the process of effective samples equalization among classes is beneficial for model training.

## 2 RELATED WORK

**Re-sampling**   Redesigned sampling frequencies for different classes are used in re-sampling based strategies. Early ideas mainly focus on under-sampling the head classes and over-sampling the tail classes. Drummond et al. (2003) argues that over-sampling is better than under-sampling because the latter process may loss important samples, while over-sampling the tail classes may lead to the over-fitting problem at the same time. Chawla et al. (2002); Han et al. (2005); He et al. (2008) By introducing the generated new data for tail through interpolation, the above problem could be solved. However, the imprecise interpolation may also introduce new noises. The process of representation learning and classifier learning should be decoupled with their suitable distributionKang et al. (2019); Zhou et al. (2020).

**Re-weighting**   Re-weighting refers to assigning different weights to loss computation denoted by the corresponding classes . The reciprocal of sample frequency is adopted to correlate with weight in early studiesHuang et al. (2016); Wang et al. (2017).Re-weighting by the number of effective samples of each class is utilized in Mahajan et al. (2018); Mikolov et al. (2013). LDAM Cao et al. (2019) adopted the loss determined by the classification decision boundary distance, where categories with larger magnitude are closer to the decision boundary. Meta-learning based methodJamal et al. (2020) also is used for a better weights estimating Zhang et al. (2021a). considers the difficulty and total number of the data to determine loss weights.In addition, some methods based on difficult samplesZhang et al. (2021a) and logits adjustmentsMenon et al. (2020) also belongs to re-weighting.

**Transfer learning**   Transfer learning attempts to transfer knowledge from source domain to enhance performance on the target domain Zhang et al. (2021b). BBN Zhou et al. (2020) is trained on the origin distribution in the early steps, which transfer to classes-balanced distribution later for the optimization of classifier. LEAP Liu et al. (2020) constructs a "feature cloud" of tail classes transferred from head ones features to better support the classification boundaries. LFME Xiang et al. (2020) trains multi-expert models separately on multiple sub-datasets, and produce a student model through knowledge distillation. RIDE Wang et al. (2020b) uses dynamic routing to control the number of experts involved.

Research for long-tailed classification mainly focuses on the above aspects. In addition, there are some theoretical studies on training strategies for long-tailed distribution. Kang et al. (2019) and Zhou et al. (2020) show an empirical law of long-tail classification research, that is, the process of representation learning and classifier learning is uncoupled. Menon et al. (2020) points out using Adam-type optimizers may not be conducive to the training for long-tailed datasets. Cui et al. (2019) introduces the concept of the effective number of samples because of the finding that the total number of non repeated samples actually participating in the training may not be as large as expected.

## 3 EFFECTIVE SAMPLING THEORY

Inspired by the concept of the effective samples Cui et al. (2019), this paper proposes a hypothesis to explain the effective sampling in training processes. We believe that the total number effective samples is the primary core factor in the representation learning, and then the next one is the utilization of effective samples between categories. The performance of the representation learning can be improved by the increasing the total number of effective samples and equalizing the effective sample utilization.

### 3.1 CONCEPT DESCRIPTION

The image information **redundancy** occurs during model training when objects have similar features. As the number of instances of a certain category increases, the probability of redundant samples usually increases, which is due to the inconsistent difficulty of data collection from source. For example of the data collection on the cat category, the total samples of hairless cats Probably much smaller than any hairy type.In addition, multiple repeated sampling from the same source for different angles suffers lower generalization performance than the separate sampling from different source in the same category. Redundancy causes the asynchronous growth of category frequency and information content.

For the sample $x_1$, $x_2$ and the encoder $f_{encoder}$, if the image substructure $s_1 \in x_1 \ and \ s_2 \in x_2$ and $\|f_{\text{encoder}}(s_1) - f_{\text{encoder}}(s_2)\| = \|z_1 - z_2\| < \delta$, then these samples are **redundant**. Then the updated conception of **effective sampling** is proposed, which refers to those sampling processes that do not generate new redundancy in the existing data sets. Specifically, for the specific structure $a_i$ and category $k$, if one sampling is performed, and the category of this sample belongs to $k$, which producing no redundancy with $a$, then an effective sampling processes happens, during which the **total number of effective samples** of the $k$ category $+1$. The above sampling process is called **effective sample sampling**. The ratio of the number of valid samples to the total number of this category is defined as the **effective sample proportion**.

Based on the concept of effective sampling and effective sample proportion, the **effective sample theory** is established. The **effective sample theory** studies how the category sampling distribution affects the actual training efficiency. In this process, we define the two concepts of the **total number of effective samples** and the **utilization rate of effective samples**, and then give the quantitative analyze for those two concepts in different sampling methods.
Suppose there are N samples in dataset with m class labels. The number of each category is $(n_1, n_2, ..., n_m)$. **The effective sample proportion** of each category is $(a_1, a_2, ..., a_m)$, and the actual **sampling frequency** is set to $(u_1, u_2, ..., u_m)$.

#### 3.1.1 SAMPLING WITH REPLACEMENT

Sampling with replacement means that the data of each category once sampled still has a certain probability to be sampled in next iteration.

**Total number of effective samples**

Let $E_{i,n}$ be the expected number of effective samples of the category $i$ sampled by $n$ times,which satisfies the following equation:

$$E_{i,n} = u_i \cdot \frac{\max(a_i n_i - E_{i,n-1}, 0)}{n_i} \cdot (E_{i,n-1} + 1) + \left(1 - u_i \cdot \frac{\max(a_i n_i - E_{i,n-1}, 0)}{n_i}\right) \cdot E_{i,n-1}$$

Simplify the above formula (see Appendix Effective Sampling Theory), and $E_{in} = n_i * a_i * (1 - (1 - \frac{u_i}{n_i})^n)$. We note that total number of effective samples of the overall dataset after sampling $n$ times is $S_n$, and we have: $S_n = \sum_{j=1}^{m} a_j n_j (1 - w_j^n)$ where $w_j = 1 - \frac{u_j}{n_j}$. When $n$ is large enough, the analytical solution of $u_i$ satisfies the following equation:

$$u_i = \frac{1 - \sum_{i \neq j} (1 - A_{ijn}) * n_j}{1 + \sum_{i \neq j} \frac{A_{ijn} * n_j}{n_i}}; A_{ijn} = \left(\frac{a_i}{a_j}\right)^{\frac{1}{n}}$$

This formula shows that the optimal sampling frequency is approximately equal to the class frequency ratio of the original distribution when the sampling times is large enough, that is $u_i \propto n_i$.

**Effective sample utilization**

The effective sample proportion is defined as $R_{i,n} = \frac{E_{i,n}}{u_i n}$, It describes the proportional of the total number of effective samples in the total number of samples of category $i$ after sampling n times. On the condition of sampling with replacement, this proportional expression is simplified as follows:

$$R_{in} = \frac{a_i n_i (1 - w_i^n)}{u_i n}$$

Consider the ratio of the effective sampling proportions $Q_{i,j}$ for any two classes i and j:

$$Q_{i,j,n} = \frac{R_{in}}{R_{jn}} = \frac{a_i * n_i * (1 - w_i^n) * u_j}{a_j * n_j * (1 - w_j^n) * u_i}$$

when $n$ is large enough, $Q_{i,j,n} = \frac{R_{in}}{R_{jn}} = \frac{a_i * n_i * u_j}{a_j * n_j * u_i}$. For sampling with replacement, the optimal sampling frequency needs to be approximately proportional to the product of the number of class i and its effective sample proportion, which is: $u_i \propto n_i * a_i$

### 3.1.2 SAMPLING WITHOUT REPLACEMENT

Sampling without replacement means that the data of each category is sampled completely according to the preset sampling frequency, which once sampled will not return to their original category set until the next epoch comes.

**Total number of effective samples**

On the condition of sampling without replacement, $E_{i,n}$ satisfies the following equation:

$$E_{in} = u_i(\min(E_{i,n-1} + a_i, a_i n_i)) + (1 - u_i)E_{i,n-1}$$

After Simplifying the above formula (see Appendix Effective Sampling Theory), $E_{in} = n * u_i * a_i$, and $S_n = \sum_{j=1}^{m} \min(a_j u_j n, a_j n_j)$.
Sort the tuple $(a_i, n_i, u_i)$ with descending order by $a_i$ and we obtain the following sequence:

$$(a_{x_1}, n_{x_1}, u_{x_1}), (a_{x_2}, n_{x_2}, u_{x_2}), \ldots, (a_{x_m}, n_{x_m}, u_{x_m})$$

When $n$ satisfies $\sum_{j=1}^{s} n_{x_j} \leq n < \sum_{j=1}^{s+1} n_{x_j}$, $S_n$ to reach its maximum value, on which condition the sampling frequency satisfies the following equation

$$u_{x_1} = \frac{n_{x_1}}{n}, \ldots, u_{x_s} = \frac{n_{x_s}}{n}, u_{x_{s+1}} = \frac{n - \sum_{j=1}^{s} n_{x_j}}{n}, u_{x_{s+2}} = 0, \ldots, u_{x_m} = 0$$

Obviously, $S_n$ just obtains the maximum value when and only when $\sum_{j=1}^{m} n_i = n = N$, where $u_i = \frac{n_i}{N}$. On the condition of sampling without replacement, the growth rate of the total number of effective samples is much greater than that on the condition of sampling with replacement. Theoretically, it is the optimal sampling strategy for increasing the total number of effective samples.

**Effective sample utilization**

The effective sample utilization $R_{i,n}$ and the ratio of the effective sampling proportions $Q_{i,j,n}$ can be expressed as follows:

$$R_{in} = \frac{\min(a_i u_i n, a_i n_i)}{u_i n}$$

$$= \begin{cases} a_i, & \text{if } n < \frac{n_i}{u_i} \\ \frac{a_i n_i}{u_i n}, & \text{otherwise} \end{cases}$$

$$Q_{i,j,n} = \frac{R_{in}}{R_{jn}} = \frac{n_i * a_i * u_j}{n_j * a_j * u_i} = 1; u_i \propto n_i * a_i$$

On the condition of sampling without replacement, a balanced utilization of effective samples among classes needs the sampling frequency $u_i$ to be proportional to he product of the number of class i and its effective sample proportion.

### 3.1.3 SOLUTION

Firstly, we found that the primary core factor affecting the training under the long-tailed distributions is the total number of effective samples. By maximizing the total number of effective samples, the encoder can obtain those gradient generated from samples with fewer redundancy information, which benefits for increasing training efficiency for the first stage of representation learning. Secondly,

as the training progresses, the actual utilization rate of a single effective sample becomes different, which lead to efficiency discrepancy of learning for different structures. The difference of those two key factors between Sampling methods will further affect the final classification accuracy.

In the previous studies, it was found that using the original sampling distribution for the first stage of training is more effective than class-balanced sampling distributions Kang et al. (2019), partly because the total number of effective samples reaches near maximum by simply setting the sampling frequency to be proportional to the sample frequency, which is well supported by our theory. However, according to the formula of maximizing the total number of effective samples and the effective sample utilization between classes, the maximization of the total number of effective samples and the total balance of effective sample utilization between categories can never be achieved theoretically the same time, due to the existence of sample redundancy. A reasonable trade-off is to ensure the total number of effective samples close to the maximization primarily and balancing the effective samples utilization between categories.

## 3.2 JITTER SAMPLING STRATEGY

The effective sampling theory reveals the contradiction between optimizing the total number of effective samples and optimizing the utilization of effective samples between categories,and Estimating the accurate redundancy of real-world category samples directly can be also difficult. Fortunately, the effective sample theory suggests that the optimal sampling frequency is actually close to the original sampling distribution, which implies that the total number effective sample can be approached as long as the distance between sampling frequency and originally distribution is in a controlled range,and the deviation from the original distribution gives a possibility to balance the utilization of effective samples. Another reasonable assumption is that, for a certain category, more sample frequency usually brings less effective sample proportion, which will be well explained in appendix (4).

Based on the above analysis, the jitter sampling strategy is proposed. We design a sampling schedule in which the sampling frequency fluctuates around the original sampling distribution, exploring to maximizing the total number of effective samples and balancing the sample utilization through random walks. For the case of sampling with replacement, we build a meta-dataloader, which contains multiple sub-dataloaders. During each iteration, it randomly selects one from meta-dataloader with a preset probability, and samples a data-batch at a sampling frequency that approximates the original distribution. For sampling without replacement, a single dataloader is used to sample a data-batch with a preset sample frequency close to the original distribution. In this process, we dynamically adjust the actual sampling distribution by introducing a control factor related to training time.

$$u_{it} \propto \ f(n_i, r); r = g(t); t : 0 \rightarrow 1$$

In the early stage of training, our strategy is relatively conservative,which adopts a sampling strategy almost same as the original distribution, and gradually explore from multiple sub-distributions as the training progresses. In the appendix, we prove that if the hyperparameters are properly chosen, jitter sampling strategy can perform better than the original strategy by trading off optimizing the above two key points.

### 3.2.1 SAMPLING WITH REPLACEMENT

On the condition of sampling with replacement, two different strategies are proposed. The first method mainly controls the change range of sampling frequency through temperature (jitter factor), where the selection probability for each sub-dataloaders are fixed. In the second method, we select the fixed original sampling distribution and the reverse sampling distribution for each sub-dataloaders, and complete the actual sampling process by dynamically adjusting the selection probabilities of the two.

(see appendix 1 for more details for the effectiveness proof process)

**Method 1**

Three sub-dataloaders are initialized with varying sampling frequencies as follows:

$$\text{dataloader}_1 : (u_1, u_2, \ldots, u_m)$$
$$\text{dataloader}_2 : (u_1^{1+\delta t}, u_2^{1+\delta t}, \ldots, u_m^{1+\delta t})$$
$$\text{dataloader}_3 : (u_1^{1-\delta t}, u_2^{1-\delta t}, \ldots, u_m^{1-\delta t})$$
$$\text{where } u_i \propto n_i$$

where $\delta t$ is jitter factor, which is updated at each epoch. The rule to update $\delta t$ is:

$$\delta t = \text{random}(0, 1) \cdot \alpha \cdot \max \left\{ \left( \frac{\text{epoch}}{\text{epoch}_{\text{total}}} \right)^{\beta}, \gamma \right\}$$

Each sampling selects one of the sub-dataloaders for sampling with a preset probability $[p_1, p_2, p_3]$. (see appendix 2 for more details for the effectiveness proof process) **Method 2**

Two sub-dataloaders are initialized accordance with:

$$\text{dataloader}_1 : (u_1, u_2, \ldots, u_m)$$
$$\text{dataloader}_2 : (u_1^{-1}, u_2^{-1}, \ldots, u_m^{-1})$$
$$\text{where } u_i \propto n_i$$

where $\delta t$ is the jitter factor, which is updated at the arrival of each epoch with the following rule:

$$\delta t = \alpha \cdot \max \left\{ \left( \frac{\text{epoch}}{\text{epoch}_{\text{total}}} \right)^{\beta}, \gamma \right\}$$

Each sampling selects one of the dataloaders for sampling with a preset probability $[1 - \delta t, \delta t]$.

The first jitter sampling method is more general, and we will demonstrate its effectiveness in detail in the appendix. The second dithering method cannot theoretically guarantee to maximize the total number of valid samples, but if it is assumed that in the real data set, the category with more sample instances has greater redundancy, then the second jittering method can be considered. For sampling without replacement, we only use a dataloader like:

$$\text{dataloader}_0 : (u_1^{1+\delta t}, u_2^{1+\delta t}, \cdots, u_m^{1+\delta t}), u_i \propto n_i$$

where $\delta t$ varies as follows:

$$\delta t = \text{random}(-0.5, 0.5) \cdot \alpha \cdot \max \left\{ \left( \frac{\text{epoch}}{\text{epoch}_{\text{total}}} \right)^{\beta}, \gamma \right\}$$

For the actual sampling, a queue is maintained separately for each category of samples, and the samples of that category are initially filled in a random order. $u_0 * N$ samples are drawn from class i for each epoch, and when the classes sample queue is emptied, a new round of filling is performed again in random order. When a new epoch arrives, the dataloader prioritizes the samples still in the queue from the previous round until all samples have been drawn. The purpose of doing this is to avoid the situation where there is a put-back sampling between each epoch.

In theory, sampling without replacement by dithering will lose valid samples, but since the probability of equalizing the utilization of valid samples will be increased at the same time, we will prove it in detail in the appendix.

## 4 EXPERIMENTS

### 4.1 EXPERIMENTAL SETUP

**Dataset** The long-tailed benchmark datasets commonly used are selected: CIFAR10-LT, CIFAR100-LT Cao et al. (2019) and ImageNet-LT Liu et al. (2019) which sampled from CIFAR10 Krizhevsky et al. (2009), CIFAR100 Krizhevsky et al. (2009) and ImageNetDeng et al. (2009), respectively. The imbalance ratio of CIFAR10-LT, CIFAR100-LT and ImageNet-LT are 100,100,256,respectively.

| Methods | top1-acc(%) |
|---|---|
| †Focal loss [Lin et al. (2017)][†] | 42.3 |
| †OLTR [Deng et al. (2009)] | 43.4 |
| †LDAM-DRW [Cao et al. (2019)] | 44.4 |
| †BBN [Zhou et al. (2020)] | 42.6 |
| †$\tau$-norm [Cao et al. (2019)] | 45.4 |
| †cRT [Cao et al. (2019)] | 42.6 |
| †LFME [Xiang et al. (2020)] | 43.8 |
| †Logit adjustment [Menon et al. (2020)] | 43.9 |
| †De-confound [Tang et al. (2020)] | 47.3 |
| †De-confound-TDE [Tang et al. (2020)] | 48.3 |
| # * RIDE(4experts+reduce) [Wang et al. (2020b)] | 49.5 |
| # * RIDE(4experts) [Wang et al. (2020b)] | 50 |
| # TLC(4experts) [Li et al. (2022)] | 49.8 |
| # J + group_norm+longtrain(**ours**) | **47.1** |
| # J + RIDE(4experts + reduce)(**ours**) | **49.5** |
| # J + RIDE (4experts)(**ours**) | **50.6** |

Table 1: evaluation results on cifar100-lt[1]

| Methods | top1-acc(%) |
|---|---|
| †Focal loss | 68.6 |
| †OLTR | 78.7 |
| †LDAM-DRW | 78.4 |
| †BBN | 42.6 |
| †$\tau$-norm | 79.6 |
| †cRT | 79.2 |
| †Logit adjustment | 81 |
| †De-confound | 72.5 |
| †De-confound-TDE | 80.4 |
| # * RIDE(4experts) | 81.7 |
| # †TLC(4experts) | 80.4 |
| # J + RIDE(3experts + reduce)(**ours**) | 81.2 |
| # J + RIDE(4experts)(**ours**) | **82** |

Table 2: evaluation results on cifar10-lt

**Evaluation Metrics** In long-tailed learning, the overall performance on all classes and the performance for head, middle and tail are usually reported. The overall performance on all classes is reported in this paper, and we average the class-specific accuracy and use the averaged accuracy as the metric.

The experiments are mainly conducted on two aspects. First, we experiment on major long-tailed classification benchmarks in the **evaluation results**, which mainly verifies the actual effectiveness of the proposed jitter sampling strategy. Taking the fairness of experiment, we follow the RIDE Wang et al. (2020b) ensemble learning framework. Second, we experiment the **ablation studies** on the effectiveness of each component.

## 4.2 EVALUATION RESULTS

**Experiments for single-experts model**

cirfa100-lt and cirfa10-lt are used as the experimental datasets, where ResNet-32 He et al. (2016) is selected as the backbone. In order to improve the classification accuracy without increasing the amount of calculation greatly, we use the a special designed module,which is called group L2norm, to expanse features before the original full connection layer of classifier through L2 normlization module by pre-designed groups. Batchsize is set to 256, and SGD is used as our optimizer. The total training epochs are 500, and the learning rate is initialized to 0.5, with a learning rate decays of 0.01 at 350 epoch and 450 epoch, respectively. The warmup epoch is set as 5. The jitter strategy setting adopts the second method on the condition of sampling with replacement. In the first stage (0-350), we set $\alpha = 1$ and $\beta = 1.5$, and in the second stage(350-500), we directly set $\delta t = 0.5$. The cross entropy loss is used as our loss function.

The module design motivation of group L2norm is to increase the richness of features, and ensure a output features with a certain controllable norm in different groups, so as to avoid the phenomenon that neural network reduces the loss by simply increasing the data norm, while the classification boundaries may not be well optimized. The disadvantage is that longer training is required when adding the norm module to the existing network module.

Table 2 shows that our proposed method (J-sampling+group L2norm+longtrain) surpasses most current methods (slightly lower than De-confound-TDE) with single backbone Resnet32, with an acceptable computational complexity increase.

**Experiments for multi-experts model**

Using the jitter sampling strategy combined with RIDE, the current sota ensemble learning methods are compared in cifar100-lt and cifar10-lt. In this group of experiments, the jitter sampling strategy on the condition of sampling replacement is adopted. The experimental settings are as follows: batchsize

---

[0]† denotes results copied from their paper, respectively.

[1]* denotes the results reported by their public code of sampling with replacement.

[2]# denotes the results reported by multi-experts

is set to 128, training epochs is set to 200, the learning rate is initially set to 0.1 with the decay rate of 0.01 at 160 epoch and 180 epoch, respectively. The warmup epoch is set to 5. The jitter setting adopts the sampling with replacement one. In the first stage (0-160), we set $\alpha = 0.05$ , $\beta = 2$ , $\gamma = 0.01$, and select cross entropy as the loss function. In the second stage, LDAM loss is adopted. Here we delete the feature enhancement method of group L2norm, because the L2norm operation plays a role of repetition with NormFC module of the classifier in the original RIDE.

It can be seen from Table 1 that on the cifar00-lt dataset, the single-model results are better than most of the experimental results (slightly lower than De-confound-TDE), and the multi-model results are better than the existing results (among which, J-sampling (ours)+RIDE(4experts) is 0.6 points higher than RIDE(4experts)). It can be seen from Table 2 that on the cifa10-lt dataset, J-sampling (ours) + RIDE (4experts) outperforms all existing algorithms.

### 4.3  ABLATION STUDIES

In this subsection, we experiment the ablation studies on the effectiveness of each component, which including replacement strategy, Jitter strategy and the training time. We further discuss two key factors that affecting the representation learning: the total number of effective samples and the effective sample utilization. The basic experimental setting keeps the same as **experiments for multi-experts model**.

| Dataset | Expert | Reduce | with(%) | without(%) |
|---------|--------|--------|---------|------------|
| CIFAR100 | 3 | 1 | 47.8 | **49** |
| CIFAR100 | 4 | 1 | 48.4 | **49.5** |

Table 3: with / without replacement

**Comparison of the replacement strategy**   3.1.1When the sampling rate is proportional to the number of class samples, the total number of effective samples without replacement is theoretically greater than the total number of effective samples with replacement. As shown in Table 3, it can be seen that the actual accuracy of sampling without replacement is significantly higher than sampling with replacement.

| Dataset | Expert | Reduce | with(%) | without(%) |
|---------|--------|--------|---------|------------|
| cifa10 | 4 | 0 | 81.7 | **82** |
| cifa100 | 3 | 1 | 47.8 | **48.7** |
| cifa100 | 3 | 0 | 49.7 | **49.8** |
| cifa100 | 4 | 1 | 48.4 | **49.5** |
| cifa100 | 4 | 0 | 50 | **50.6** |
| ImageNet | 3 | 1 | 54 | **54.1** |
| ImageNet | 3 | 0 | **54.6** | 54.5 |
| ImageNet | 4 | 1 | 54.6 | **54.9** |
| ImageNet | 4 | 0 | 55 | **55.2** |

Table 4: with / without jitter

**Comparison of the Jitter strategy**   We compare the jitter with and without replacement sampling on cirfa100 and imagent-lt, respectively. We have proved the effect of jitter in appendix in theory, which is also validated from the experimental, that adding jitter within sampling frequency can actually improve the accuracy with a certain probability. For non-replacement sampling, although the total number of effective samples theoretically decreases slightly, at the same time, the utilization of effective samples is also balanced. One of a possible reason for why there is no obvious accuracy improvement on imagenet-lt,compared to cirfa100-lt,is that the image size of imagenet-lt is larger than the cifar series dataset, so the redundancy between images is not high (the foreground only occupies part of the image, while the background difference between instances is obvious).

| Dataset | Expert | Reduce | with replacement + jitter | | without replacement + no jitter | |
|---|---|---|---|---|---|---|
| | | | short train | long train | short train | long train |
| cifa100 | 3 | 1 | 48.7 | **50** | 49 | - |
| cifa100 | 4 | 1 | 49.5 | **50.4** | 49.5 | - |
| cifa100 | 4 | 0 | 50.6 | **51.5** | - | - |
| ImageNet | 3 | 1 | - | **55.2** | 54 | 55.2 |
| ImageNet | 3 | 0 | - | **55.2** | - | 55 |
| ImageNet | 4 | 1 | - | **55.8** | 54.6 | 56 |

Table 5: short train

**Comparison of the training time**    The training epochs is positively correlated with the total number of effective samples. When training epochs is normal, it is obvious that the total number of effective samples on the condition of sampling with replacement is lower than the without one. Jitter strategy helps increasing the total number of effective samples and balance the effective sample utilization between categories. For longer training, the effective sample has been saturated, leading to a limited accuracy improvement.

## 5    CONCLUSION

We have established an effective sampling theory to explain the sampling efficiency gap in different sampling methods and a jitter sampling strategy is developed to improve the actual training effect,based on which our proposed methods perform well on many long-tailed datasets.If we can find a way to eliminate information redundancy precisely, our theory may be further optimized. We will explore it in the next experiment.

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

## A  APPENDIX

### A.1  APPENDIX 1

Jittering method in Sampling with replacement is effective.

**Proof 1**: In sampling with replacement, the total number of effective samples obtained using jittering method 1 is greater than the that of the original sampling with replacement, and the effective sample utilization is equalized between classes.

Let the expectation of the total number of effective samples sampled by the jittering method one be $J_n$, whose values are:

$$
\begin{aligned}
J_n &= E(S_n) \\
&= \sum_{i=1}^{3} \left\{ p_i \sum_{j=1}^{m} a_j n_j \left( 1 - \left( 1 - \frac{u_j^{(i)}}{n_j} \right)^n \right) \right\} \\
&= p_1 G(1) + p_2 G(1 + \delta t) + p_3 G(1 - \delta t)
\end{aligned}
$$

Where $u_j^{(i)}$ is the sampling rate of the i-th dataloader for the j-th class. Let $u_j(x)$ is a function that calculates the sampling rate of the j-th class in dataloaders. The sampling rate of dataloader 1 is $u_j^{(1)} = u_j(1)$. The sampling rate of dataloader 2 is $u_j^{(2)} = u_j(1 + \delta t)$. The sampling rate of dataloader 1 is $u_j^{(3)} = u_j(1 - \delta t)$.

$$
u_j(x) = \frac{n_j^x}{\sum_{k=1}^{m} n_j^x}
$$

And $G(x)$ is a function with:

$$
\begin{aligned}
G(x) &= \sum_{j=1}^{m} a_j n_j \left( 1 - \left( 1 - \frac{u_j(x)}{n_j} \right)^n \right) \\
&= \text{Const} - \sum_{j=1}^{m} a_j n_j \left( 1 - \frac{u_j(x)}{n_j} \right)^n
\end{aligned}
$$

Consider the derivative of G with respect to x:

$$\frac{\partial G}{\partial x} = -\sum_{j=1}^{m} a_j n_j \cdot n \left(1 - \frac{u_j(x)}{n_j}\right)^{n-1} \cdot \left(-\frac{1}{n_j}\right) \cdot \frac{\partial u_j(x)}{\partial x}$$

$$= \sum_{j=1}^{m} \frac{a_j}{n} \left(1 - \frac{u_j(x)}{n_j}\right)^{n-1} \cdot \frac{n_j^x \ln n_j \sum_{k=1}^{m} n_k^x - n_j^x \sum_{k=1}^{m} (n_k^x \ln n_k)}{\left(\sum_{k=1}^{m} n_k^x\right)^2}$$

When $x = 1$:

$$\left.\frac{\partial G}{\partial x}\right|_{x=1} = n \left(1 - \frac{1}{N}\right)^{n-1} \cdot \sum_{j=1}^{m} a_j \cdot \frac{n_j \ln n_j \sum_{k=1}^{m} n_k - n_j \sum_{k=1}^{m} (n_k \ln n_k)}{\left(\sum_{k=1}^{m} n_k\right)^2}$$

If $\left.\frac{\partial G}{\partial x}\right|_{x=1} = 0$, there is:

$$\sum_{j=1}^{m} \left\{ a_j n_j \ln n_j \sum_{k=1}^{m} n_k \right\} = \sum_{j=1}^{m} \left\{ a_j n_j \sum_{k=1}^{m} (n_k \ln n_k) \right\}$$

This usually doesn't hold. It is worth mentioning that when $a_1 = a_2 = \cdots = a_m$, that is, when the redundancy of each class of the dataset is the same, the above equation holds.

Therefor, normally:

$$\left.\frac{\partial G}{\partial x}\right|_{x=1} > 0 \text{ or } \left.\frac{\partial G}{\partial x}\right|_{x=1} < 0$$

Then there must exist $\delta t$ such that $G(1 + \delta t) > G(1)$ or $G(1 - \delta t) > G(1)$. So, a probability combination $(p_1, p_2, p_3)$ must exist that let $p_1 G(1) + p_2 G(1 + \delta t) + p_3 G(1 - \delta t) > G(1)$.

In summary, we prove that there must exist parameters $\delta t$ and $(p_1, p_2, p_3)$ that allow the the total number of effective samples obtained using jittering method one of sampling with replacement is greater than the total number of effective samples.

When n is sufficiently large, the effective sample utilization between classes is the same as the formula when there is sampling without replacement. The sample utilization between classes is balanced at this time, as we will prove in Appendix 2.

**Proof 2**: In sampling with replacement, the utilization of the effective samples between classes obtained by jittering method 2 is more balanced than that of the original sampling with replacement.

The effective sample utilization for the ith category is $R_{i,n}$. In method 2 of sampling with replacement, when $n$ is large enough, it is:

$$R_{i,n} = x \cdot \frac{a_i n_i (1 - w_i^n)}{u_i n} + (1 - x) \cdot \frac{a_i n_i (1 - \overline{w_i}^n)}{\overline{u_i} n}$$

$$\approx x \cdot \frac{a_i n_i}{u_i n} + (1 - x) \cdot \frac{a_i n_i}{\overline{u_i} n}$$

Where $w_i = 1 - \frac{u_i}{n_i}, \overline{w_i} = 1 - \frac{\overline{u_i}}{n_i}, \overline{u_i} = \frac{u_i^{-1}}{\sum_{j=1}^{m} u_j^{-1}}$.

The derivative of $R_{i,n}$ is:

$$\frac{\partial R_{i,n}}{\partial x} = \frac{a_i n_i}{u_i n} - \frac{a_i n_i}{\overline{u_i} n}$$

A more balanced effective sample utilization corresponds to a distribution of category effective sample utilization $P = \{\frac{R_{1,n}}{\sum_k R_{k,n}}, \ldots, \frac{R_{m,n}}{\sum_k R_{k,n}}\}$ and a uniform distribution $Q = \{\underbrace{\frac{1}{m}, \ldots, \frac{1}{m}}_{m}\}$ with

less KL divergence.

$$\mathrm{KL}(Q\|P) = \sum_{i=1}^{m} Q_i \log \frac{Q_i}{P_i}$$

$$= -\log m - \frac{1}{m}\sum_{i=1}^{m}\log R_{i,n} + \log\left(\sum_{i=1}^{n} R_{i,n}\right)$$

Consider the derivative of the KL divergence.

$$\frac{\partial \mathrm{KL}}{\partial x} = -\frac{1}{m}\sum_{i=1}^{m}\frac{\frac{\partial R_{i,n}}{\partial x}}{R_{i,n}} + \frac{\sum_{i=1}^{m}\frac{\partial R_{i,n}}{\partial x}}{\sum_{i=1}^{m} R_{i,n}}$$

When $x = 1$, $R_{i,n} = \frac{a_i n_i}{u_i n}$.

$$\frac{\partial \mathrm{KL}}{\partial x}\bigg|_{x=1} = -\frac{1}{m}\sum_{i=1}^{m}\frac{\frac{a_i n_i}{u_i n} - \frac{a_i n_i}{\bar{u}_i n}}{\frac{a_i n_i}{u_i n}} + \frac{\sum_{i=1}^{m}\frac{a_i n_i}{u_i n} - \frac{a_i n_i}{\bar{u}_i n}}{\sum_{i=1}^{m}\frac{a_i n_i}{u_i n}}$$

$$= \frac{1}{m}\cdot\sum_{i=1}^{m}\frac{u_i}{\bar{u}_i} - \frac{\sum_{j=1}^{m} a_j \cdot \frac{u_j}{\bar{u}_j}}{\sum_{j=1}^{m}\frac{a-j n_j}{u_j n}}$$

$$= \left(\sum_{k=1}^{m} u_k^{-1}\right)\cdot\left(\frac{1}{m}\sum_{i=1}^{m} u_i^2 - \frac{\sum_{i=1}^{m} a_i u_i^2}{\sum_{j=1}^{m} a_i}\right)$$

From Appendix 4, the proportion of effective samples is low for categories with large sample size, which means that the lager $u_i$ is, the smaller $a_i$ is. Therefore,

$$\frac{1}{m}\sum_{i=1}^{m} u_i^2 - \frac{\sum_{i=1}^{m} a_i u_i^2}{\sum_{j=1}^{m} a_i} > 0$$

That means

$$\frac{\partial \mathrm{KL}}{\partial x}\bigg|_{x=1} > 0$$

Therefore, there exists $\delta t \in (0,1)$ such that $\mathrm{KL}(1 - \delta t) < \mathrm{KL}(1)$. In other words, the dithering method 2 can make the effective sample utilization more balanced.

## A.2  APPENDIX 2

Jittering method in Sampling without replacement is effective.

**Proof**: In sampling without replacement, the effective sample utilization of a jittering method would be more balanced than without jittering.

The effective sample utilization for the ith class is:

$$R_{i,n} = \frac{a_i n_i}{u_i(x)\,n}$$

A more balanced effective sample utilization corresponds to a distribution of category effective sample utilization $P = \{\frac{R_{1,n}}{\sum_k R_{k,n}}, \ldots, \frac{R_{m,n}}{\sum_k R_{k,n}}\}$ and a uniform distribution $Q = \{\underbrace{\frac{1}{m}, \ldots, \frac{1}{m}}_{m}\}$ with

less KL divergence.

$$\text{KL}(Q\|P) = \sum_{i=1}^{m} Q_i \log \frac{Q_i}{P_i}$$

$$= \frac{1}{m} \sum_{i=1}^{m} \log \frac{1}{mP_i}$$

$$= -\log m - \frac{1}{m} \sum_{i=1}^{m} \log P_i$$

Consider the sampling rate function $u_i(x)$.

$$u_i(x) = \frac{n_i^x}{\sum_{k=1}^{m} n_k^x}$$

For $x = 1 + \delta t$ and $\delta t > 0$, the sampling rate increases for the head classes and decreases for the tail classes. We have $n < \frac{n_i}{u_i(x)}$ in tail classes. So, effective sample utilization of tail classes is $R_{i,n} = a_i$, and effective sample utilization of tail classes is $R_{i,n} = \frac{a_i n_i}{u_i(x)n}$.

Might as well let m-th class has the largest number of samples. There exists $\delta t$ quite small such that $R_{m,n} = \frac{a_m n_m}{u_m(x)n}$. Ant the utilization of the others is $a_i$. So the KL divergence is:

$$\text{KL}(Q\|P) = -\log m - \frac{1}{m} \left( \sum_{i=1}^{m-1} \log \frac{a_i}{\sum_{k=1}^{m-1} a_k + \frac{a_m n_m}{u_m(x)n}} + \log \frac{\frac{a_m n_m}{u_m(x)n}}{\sum_{k=1}^{m-1} a_k + \frac{a_m n_m}{u_m(x)n}} \right)$$

$$= -\log m - \frac{1}{m} \left( \sum_{i=1}^{m-1} \log a_i + \log \frac{a_m n_m}{u_m(x)n} - m \log \left( \sum_{k=1}^{m-1} a_k + \frac{a_m n_m}{u_m(x)n} \right) \right)$$

The derivative of KL divergence with respect to x is:

$$\frac{\partial \text{KL}}{\partial x} = -\frac{1}{m} \cdot \left( -\frac{u_m(x)n}{a_m n_m} \cdot \frac{a_m n_m}{u_m^2(x)n} \cdot \frac{\partial u_m(x)}{\partial x} + m \cdot \frac{1}{\sum_{k=1}^{m-1} a_k + \frac{a_m n_m}{u_m(x)n}} \cdot \frac{a_m n_m}{u_m^2(x)n} \cdot \frac{\partial u_m(x)}{\partial x} \right)$$

$$= \frac{1}{m} \cdot \left( \frac{u_m(x)n}{a_m n_m} - \frac{m}{\sum_{k=1}^{m-1} a_k + \frac{a_m n_m}{u_m(x)n}} \right) \cdot \frac{a_m n_m}{u_m^2(x)n} \cdot \frac{\partial u_m(x)}{\partial x}$$

Where $\frac{\partial u_m(x)}{\partial x}$ is:

$$\frac{\partial u_m(x)}{\partial x} = \frac{n_i^x \ln n_i \cdot \sum_{k=1}^{m} n_k^x - n_i^x \sum_{k=1}^{m} (n_k^x \ln n_k)}{\left( \sum_{k=1}^{m} n_k^x \right)^2}$$

When $x = 1$, we have:

$$\frac{\partial \text{KL}}{\partial x} \bigg|_{x=1} = \frac{1}{m} \cdot \left( \frac{1}{a_m} - \frac{m}{\sum_{k=1}^{m} a_k} \right) \cdot \frac{a_m N}{n_m} \cdot \frac{n_m \ln n_m \cdot \sum_{k=1}^{m} n_k - n_m \cdot \sum_{k=1}^{m} (n_k \ln n_k)}{\left( \sum_{k=1}^{m} n_k \right)^2}$$

Since $n_m$ is the largest and $a_m$ is the smallest from Appendix 4, we have

$$\frac{\partial \text{KL}}{\partial x} \bigg|_{x=1} < 0$$

So when $\delta t > 0$ and closer to 0, the jittering of $u_i(1 + \delta t)$ causes $\text{KL}(Q\|P)$ to drop, that is, the effective sample utilization between classes is more balanced.

Similarly, it can be shown that when $\delta t > 0$ and closer to 0, the jittering of $u_i(1 - \delta t)$ causes the effective sample utilization between classes be more balanced.

## A.3 APPENDIX 3

### A.3.1 SAMPLING WITH REPLACEMENT

**Total number of efficient samples**

Define the effective number of samples obtained for the $i$-th class after $n$ sampling as $E_{i,n}$, whose recursive formula is

$$E_{in} = u_i \cdot \frac{\max(a_i n_i - E_{i,n-1}, 0)}{n_i} \cdot (E_{i,n-1} + 1) + \left(1 - u_i \cdot \frac{\max(a_i n_i - E_{i,n-1}, 0)}{n_i}\right) \cdot E_{i,n-1}$$

When the total number of effective samples reaches $a_i n_i$, no new effective samples are added. Therefore, to simplify the discussion, we consider the case where the upper bound has not yet been reached. Simplifying the above equation yields:

$$E_{in} = a_i u_i + \left(1 - \frac{u_i}{n_i}\right) E_{i,n-1}$$

Let $w_i = 1 - \frac{u_i}{n_i}$, it's easy to know:

$$\frac{E_{in}}{w_i^n} = \frac{a_i u_i}{w_i^n} + \frac{E_{i,n-1}}{w_i^{n-1}}$$

$$\frac{E_{in}}{w_i^n} = \sum_{j=1}^{n} \frac{a_i u_i}{w_i^j} + E_0$$

$$E_{in} = a_i u_i \cdot \frac{w_i^n - 1}{w_i - 1}$$

$$= a_i n_i (1 - w_i^n)$$

We note that total number of effective samples of the overall dataset after sampling $n$ times is $S_n$, and we have:

$$S_n = \sum_{j=1}^{m} a_j n_j (1 - w_j^n)$$

Maximize $S_n$:

$$\max \quad S_n$$

$$\text{s.t.} \quad \sum_{i=1}^{m} u_i = 1$$

$$u_i > 0$$

We introduce Lagrange multipliers and try to solve for the conditions satisfied when $S_n$ reaches its extreme value:

$$L(u_1, \ldots, u_m, \lambda) = S_n + \lambda \left(1 - \sum_{i=1}^{m} u_i\right)$$

Calculate its derivative:

$$\frac{\partial L}{\partial u_i} = -a_i n \left(1 - \frac{u_i}{n_i}\right)^{n-1} - \lambda$$

Let $\frac{\partial L}{\partial u_i} = 0$, get:

$$\frac{\partial L}{\partial u_i} = \frac{\partial L}{\partial u_j}$$

$$\left(\frac{a_i}{a_j}\right)^{1/n} = \frac{1 - \frac{u_j}{n_j}}{1 - \frac{u_i}{n_i}}$$

When $n$ is large enough, the analytic solution of $u_i$ satisfies the following equation.

$$u_i = \frac{1 - \sum_{i \neq j} n_j (1 - A_{i,j,n})}{1 + \sum_{i \neq j} n_j A_{i,j,n}/n_i}$$

$$A_{i,j,n} = \left( \frac{a_i}{a_j} \right)^{\frac{1}{n}}$$

This equation shows that in sampling with replacement the optimal sampling frequency is approximately equal to the category frequency ratio of the original distribution when the number of samples is large enough, which also implies that we can theoretically obtain close to the upper limit of the total number of valid samples by using a sampling ratio that approximates the original distribution. The optimal sampling rate $u_i$ satisfies:

$$u_i \propto n_i$$

**Effective sample utilization**

The effective sample proportion is defined as follows.

$$R_{i,n} = \frac{E_{i,n}}{u_i n}$$

In sampling with replacement, this expression is simplified as follows:

$$R_{i,n} = \frac{a_i n_i (1 - w_i^n)}{u_i n}, \quad \text{where } w_i = 1 - \frac{u_i}{n_i}$$

Consider the ratio of the effective sampling proportions of any two classes $Q_{i,j}$.

$$Q_{i,j} = \frac{R_{in}}{R_{jn}}$$
$$= \frac{a_i n_i u_j (1 - w_i^n)}{a_j n_j u_i (1 - w_j^n)}$$

It is not difficult to find that when $n$ is sufficiently large, $Q_{i,j}$ satisfies:

$$Q_{i,j} = \frac{a_i n_i \cdot u_j}{a_j n_j \cdot u_i}$$

Therefore, in sampling with replacement, when the sampling frequency approximates the effective number of the class is proportional, the effective sample utilization is balanced. That is:

$$u_i \propto a_i n_i$$

### A.3.2   SAMPLING WITHOUT REPLACEMENT

**Total number of effective samples**

In sampling without replacement, $E_{in}$ satisfies the following relation.

$$E_{i,n} = u_i (\min(E_{i,n-1} + a_i \cdot 1, a_i n_i)) + (1 - u_i) E_{i,n-1}$$

Simplify to get:

$$E_{i,n} = a_i u_i n$$

Therefore $S_n$ satisfies:

$$S_n = \sum_{j=1}^{m} \min(a_j u_j n, a_j n_j)$$

May wish to consider that $a_1 \leq a_2 \leq \cdots \leq a_m$. For any $k \in [1, m]$, when $n$ satisfies:

$$\sum_{i=1}^{k} n_i \leq n < \sum_{i=1}^{k+1} n_i$$

The condition for $S_n$ to reach its extreme value is

$$u_1 = \frac{n_1}{n}, \ \ldots, u_k = \frac{n_k}{n}, \ u_{k+1} = \frac{n - \sum_{j=1}^{k} n_j}{n}, \ u_{k+2} = 0, \ \ldots, \ u_m = 0$$

It is easy to know that $S_n$ just obtains the maximum value of the above equation when and only when $\sum_{j=1}^{m} n_i = n = N$, when it only needs to satisfy:

$$u_i = \frac{n_i}{N}$$

**Effective sample utilization**

Define after $n$ samples the effective sample utilization $R_{in}$ as follows.

$$R_{in} = \frac{\min(a_i u_i n, a_i n_i)}{u_i n}$$

$$= \begin{cases} a_i, & \text{if } n < \frac{n_i}{u_i} \\ \frac{a_i n_i}{u_i n}, & \text{otherwise} \end{cases}$$

When $n$ is large enough, for any $i$, $j$, considering the condition of 1, there are:

$$Q_{i,j,n} = \frac{R_{in}}{R_{jn}}$$

$$= \frac{a_i n_i \cdot u_j}{a_j n_j \cdot u_i} = 1$$

For sampling without replacement, the condition for achieving a balanced utilization of effective samples among classes is that the sampling frequency must be proportional to the number of effective samples in a class.

$$u_i \propto a_i n_i$$

### A.4 Appendix 4:

**Proof**: When the number of classes is sufficiently large, the classes redundancy and the number of classes are negatively correlated.

We assume that class $i$ obeys a priori Gaussian distribution, and the actual data of class i is actually obtained from that Gaussian distribution.

Each point on the numerical axis represents the sample we actually sampled, and the probability density of the location it was actually sampled is

$$f(t) = \frac{1}{\sigma\sqrt{2\pi}} \exp\left(-\frac{(t - \mu_i)^2}{\sigma^2}\right)$$

We define two samples $t_i, t_j$ as redundant if their sample positions $|t_i - t_j| < \delta$. Let the sampling position $t_0$, then the probability that the sampling position is greater than position $t_0$ is $P = P(t > t_0)$.

$$P(t > t_0) = 1 - \int_{-\infty}^{t_0} f(t) \mathrm{d}t$$

The expectation of the number the sample location is greater than $t_0$ is $NP$ for $N$ independent acquisitions times.

Consider the case when $N * P = 1$, whose physical meaning is that after collecting $N$ times expects only once its sampling position is larger than $t_0$. $t_0$ can be considered as the average upper bound of the sample position of N times sampling. Considering symmetry, the lower bound of the sampling position is $2\mu_i - t_0$. Then the upper level of the effective sample proportion $\overline{a_i}$ satisfies:

$$\overline{a_i} = \frac{2t_0 - 2\mu_i}{\sigma N}$$

It's easy to know:

$$\lim_{t_0 \to +\infty} \overline{a_i} = \frac{2f(t_0)(t_0 - u_i)^2}{\delta} = 0$$

Although a rigorous derivation of $a_i$ cannot be given, we show that the upper bound $a_i$, $\overline{a_i}$, decreases monotonically after sampling a certain range and $\overline{a_i}$ tends to 0 when the number of samples tends to infinity. Since then, we have completed the derivation of the negative correlation between the category redundancy and the number of class samples.

The above is based on the assumption that the dimension sample is 1, while the actual datatype we deal with is much more complicated like image-type. In the real case, we think that the an image sample $I_i$ can be represented as its potential variable $X_i$ which can be generated by a self encoding network,that is:

$$f_{encoder}(I_i) = [X_{i1}, X_{i2}, X_{i3}, ....X_{is}] = X_i; f_{decoder}(X_i) = I_i$$

In the study of VAE, $X_i$is defined as the potential variable obeying a specific Gaussian distribution:

$$X_i \sim N(u_i, \sigma^2)$$

and then the $a_i$ of $I_i$ can be expressed as follows:

$$a_i \leq a_{1i} * a_{2i} * a_{3i}... * a_{si}$$

$a_{si}$ donates the effctive sample proposition of $X_i$,and when $X_i$ maintains statistical independence with each other($X_j$), equality can be established.

