# OpenReview forum: "HOW SAMPLING AFFECTS TRAINING: AN EFFECTIVE SAMPLING THEORY STUDY FOR LONG-TAILED IMAGE CLASSIFICATION"
_ICLR.cc/2023/Conference — Submitted to ICLR 2023_

### Official Review · Reviewer_y4qK · 2022-10-21

**Confidence:** 5
**Correctness:** 2
**Technical Novelty And Significance:** 2
**Empirical Novelty And Significance:** 2
**Recommendation:** 1

**Clarity, Quality, Novelty And Reproducibility:**

Severe issues and limitations regarding clarity, quality, novelty and reproducibility.

**Strength And Weaknesses:**

Strengths:

- Effective number of samples is a promising idea to explore for sampling.

Weaknesses:

1. The paper is extremely difficult to read, making it infeasible to comprehend its contributions, its line of reasoning, its methodology, its overlaps / differences to prior work etc.

1.1 The paper has too many grammatical errors, some which are listed below for the first page. I gave up providing recommendations for the rest of the paper. I strongly suggest the authors to submit only polished, proof-read papers to venues, especially top ones such as ICLR. In its current state, the papers feels like it has been rushed and it might be containing different types of errors, not only grammatical ones.

1.2. Many unclear parts or statements, e.g. (only for the first one/two pages):

- "inconsistent difficulty of data collection from source" => It is not clear what's meant by inconsistent difficulty.
- "For example of the data collection on the cat category, the total samples of hairless cats Probably much smaller than any hairy type" => This is Probably not a convincing statement.
- "multiple repeated sampling from the same source for different angles" => What is an angle here?

- It is not clear how the formulations in Section 3.1.1 are derived.

- Table 1, 2: It is not clear what the imbalance ratios are.

2. I find it too strong to call an approach a theory.

3. Section 2 is missing approaches using regularization or normalization as a form of regularization, e.g. "Long-Tailed Recognition via Weight Balancing", CVPR2022. The paper also missed earlier work focusing on the design of sampling approaches: "Assessing the Effect of Training Sampling Design on the Performance of Machine Learning Classifiers for Land Cover Mapping Using Multi-Temporal Remote Sensing Data and Google Earth Engine", 2021.

4. Another important limitation is about evaluation. The paper provides a mixed comparison in Tables 1 & 2, without discussing explicitly the contribution over existing sampling approaches. The paper should dedicate a separate section & analysis for a comparison with existing sampling approaches.

5. The paper provides a SOTA comparison only on CIFAR10 & CIFAR100. Results on more challenging datasets such as ImageNet-LT and iNaturalist are necessary to see how the method fairs in challenging settings.

Minor comments:

- "This phenomena dose not only occurs in image classification" => "This phenomena dose not only occur in image classification".
- "semantic segmentation He et al. (2021); Wang et al. (2020a)" => I think the correct cite command is \citep. The correct form should have been: "semantic segmentation (He et al., 2021; Wang et al., 2020a)". Similar problems exist in the rest of the paper.
- "Kang et al. (2019); Zhou et al. (2020) mentions" => "Kang et al. (2019); Zhou et al. (2020) mention".
- "two stages learning" => "two-stage learning".
- "Sampling process need be conducted" => "Sampling process needs be conducted".
- "*" means multiplication in a programming language. You should use \cdot or \times.
- Please follow the following guide for writing equations: http://www.ai.mit.edu/courses/6.899/papers/mermin.pdf


**Summary Of The Paper:**

In this paper, the authors study the problem of long-tailed visual recognition. For this end, they propose a new sampling strategy from the perspective of "effective samples" proposed earlier by Cui et al. (2019).

**Summary Of The Review:**

Severe issues and limitations regarding clarity, quality, novelty and reproducibility.

---

### Official Review · Reviewer_3mbq · 2022-10-25

**Confidence:** 4
**Correctness:** 3
**Technical Novelty And Significance:** 1
**Empirical Novelty And Significance:** Not applicable
**Recommendation:** 3

**Clarity, Quality, Novelty And Reproducibility:**

This paper presents a new theory, but as the main contribution, theory analysis is insufficient, and the experimental results do not support its effectiveness. In addition, some claims in this paper are not further justified. The details and code of the proposed method are not given, which limits reproducibility.


**Strength And Weaknesses:**

This paper focuses on a fundamental and valuable problem, the long-tail problem on real-world datasets. As one of the solutions to the long-tailed classification problem, it is significant to study re-sampling, especially the theoretical analysis of sampling strategies on long-tailed distributions.
Several experiments are carried out on three real-world image datasets. The classification performances of the three proposed methods are evaluated by comparing them with thirteen baseline methods (both single-expert and multi-expert). In addition, the authors conduct ablation studies to highlight the importance of each component in the proposed methods.
However, my main concern is that the novelty and significance of the new theory are not well stated and that the experimental results do not support the contributions well. Here are the detailed comments:
[Presentation] First of all, the writing of the paper is not easy to follow for two reasons. (1) There are many statements and claims without clear meanings. For example, the meaning of the actual sampling frequency may need to explain when first mentioned. (2) Some notations are not clearly defined or have been used before the definition. For example, the definition of the subscript "m" is not given or is written in an inconspicuous place.
[Novelty] I think the proposed new theory is not innovative enough compared to the existing conclusions (Kang et al., Cui et al.), and it does not bring a conceptual breakthrough. According to the statements in the paper, I cannot understand what and how the new theory improves the existing theories on which it is based. Currently in my opinion, it simply expresses an intuitive conclusion in mathematical terminology and may not bring a few new perspectives on how to design sampling methods in the future.
[Method Details] Method 1 and Method 2 lack important details. I cannot find any description of the parameter settings. For example, what do \alpha, \beta, and \gamma represent? Are they learnable, or are they hyperparameters? These are critical parts of the methods, and the lack of information hampers my understanding of how the methods work.
[Experiments] In the experimental results of image classification, the methods proposed by the paper show equivalent or even reduced performances to some of the other alternative methods. Based on such experimental results, contributions compared to previous work cannot be significantly proved. In addition, some parts of the manuscript (abstract, conclusion) seem to overstate the performance and should be reformulated.


**Summary Of The Paper:**

This paper focuses on classification problems for long-tailed image datasets. Specifically, the authors first establish an effective sampling theory to theoretical study the total number of effective samples and effective sample utilization in sampling with/without replacement. Then two general jitter sampling strategies are proposed for increasing the actual training effect in sampling with replacement. Finally, the experiments on three long-tailed datasets for many baselines indicate the classification performance of the proposed methods.


**Summary Of The Review:**

This paper notices a critical problem, the long-tail distribution in image classification. New sampling strategies are proposed to address this problem. However, the novelty of the technical contribution is lacking, and the experimental results do not support the claim that the performance is improved.

---

### Official Review · Reviewer_xj1Z · 2022-10-30

**Confidence:** 5
**Correctness:** 2
**Technical Novelty And Significance:** 2
**Empirical Novelty And Significance:** 2
**Recommendation:** 3

**Clarity, Quality, Novelty And Reproducibility:**

Overall the paper is easy to read and follow. Many typos in Section 3.1 and 3.2 e.g “...the total samples of hairless cats Probably…”, “...hair type.\space\In addition…”, “For the sample(s) x1, x2, and the …”.

**Strength And Weaknesses:**

Strengths:
- The paper provides a theoretical explanation for the decoupling representation and classifier approach proposed by Kang et al [1].
- A novel sampling strategy is proposed.
- Experiments are conducted on long-tail learning benchmarks to show the effectiveness of the sampling strategy.

Weaknesses:
- The provided results do not show the effectiveness of the proposed method.
  	(1) Results for group_norm + longtrain (without jitter strategy) is missing. While
                 “J+group_norm + longtrain” achieves 47.1 on cifar-100, it does not show the
                 effectiveness of the jitter strategy.
	(2) In addition, the accuracy for both “J+RIDE [0] (4 experts+reduce)” and “RIDE [2]
                 (4 experts+reduce)” is 49.5 (no improvement is shown).
- Many, Med, and Few shot accuracies are missing. They are important to understand the effectiveness of the proposed approach compared to other methods (sampling and non-sampling methods)..
- The paper proposes a jitter sampling strategy, based on effective theory, which aims to select effective samples to improve training.
	(1) There are no comparisons to other sampling approaches like [3] or [4].
	(2) Active learning approaches aim for the same task: Select samples that are most
                 effective for training. Comparisons to such methods (e.g [5]) would
                 be sensible.
- The performance improvement of the proposed approach compared to “RIDE [2]”  is very small.  +0.6% on cifar100, +0.3 on cifar10. Although the improvement can be due to the new sampling approach, it can also be due to variance. Reporting results variance for the proposed method and baselines is necessary.
- The approach is only effective on small image-size datasets. The paper states that no obvious accuracy can be seen on Imagenet-lt dataset due to large image size. No empirical results on Places-LT and iNatrualist.


[1] Decoupling Representation and Classifier for Long-Tailed Recognition, Kang et al.
[2] Long-tailed Recognition by Routing Diverse Distribution-Aware Experts, Wang et al.
[3] The Effects of Data Sampling with Deep Learning and Highly Imbalanced Big Data,
[4] Not All Samples Are Created Equal: Deep Learning with Importance Sampling, Katharopoulos et al.
[5] Active Learning for Imbalanced Datasets, Aggrawal et al.


**Summary Of The Paper:**

The paper proposes an “effective sampling” theory which provides a theoretical explanation for the two-stage training scheme utilized in long-tail learning. The paper claims that both the total number of effective samples and the effective sample utilization affect training and performance for long-tail datasets. A general jitter sampling strategy is proposed based on the theory. Experiments are shown to verify the approach by comparing it to SOTA long-tail approach on common long-tail benchmarks.

**Summary Of The Review:**

Leaning to reject the paper for lack of empirical support, missing important baselines, and not being applicable for large-size datasets.

---

### Official Review · Reviewer_vZeb · 2022-10-31

**Confidence:** 3
**Correctness:** 2
**Technical Novelty And Significance:** 2
**Empirical Novelty And Significance:** 2
**Recommendation:** 3

**Clarity, Quality, Novelty And Reproducibility:**

The paper is not well written, both its clarity and quality need to be improved.

The efficient sampling methods sound interesting, but whether the method is novel compared to existing methods is hard to justify. The authors are suggested to compare their method to existing sampling strategies to highlight their contributions.

No code to verify the reproducibility.

**Strength And Weaknesses:**

Cons:

1. This paper is based on the observation that "data redundancy hurts long-tailed image classification performance" in Section 3. Is there any statistical or theoretical evidence on this point? This point needs to be better justified with evidence, otherwise, the proposed methods will be vacuous.

2. The presentation in Section 3 needs to be improved. The first 3 paragraphs in Section 3.1 is hard to follow and the definitions are mixed together. Maybe consider formally defining "effective sampling", "effective samples", and "effective sample proportion". Besides, how do you select substructures s1 s2 from image x1 x2? How are the effective samplings selected from the full dataset? How a1 ... am are computed? Many details are missing when reading this section.

3. Could you please explain why "the maximization of the total number of effective samples and the total balance of effective sample utilization between categories can never be achieved theoretically at the same time, due to the existence of sample redundancy." above Section 3.2?

4. The authors need to highlight the difference between the effective sampling method with existing re-sampling and re-weighting methods.

5. Experiment section is also hard to follow. The authors are suggested to briefly introduce the experiment setup, highlight the key results of the experiment, then provide a discussion and take-home messages for readers.

**Summary Of The Paper:**

This paper studies long-tailed image classification problem. The authors identify two key factors that affect the performance of long-tailed image classification: (1) the total number of effective samples  and (2) the effective sample utilization. Based on this finding, the authors proposes an effective sampling theory for theoretical explanation and propose a general jitter sampling strategy for practical implementation.


**Summary Of The Review:**

In general, I found the efficient sampling idea interesting. However, the motivation for this efficient sampling is not strong enough, which requires either statical or theoretical evidence.

The Paper is hard to read. There are many types and the paper's organization in Sections 3 and 4 might also need to be improved.

The method section is not very informative, and details including how effective samples are selected are not carefully introduced.

---

### Decision · Program_Chairs · 2023-01-20

**Decision:**

Reject

**Justification For Why Not Higher Score:**

All reviewers recommended reject and pointed major issues with theory and experiments.

**Justification For Why Not Lower Score:**

N/A

**Metareview: Summary, Strengths And Weaknesses:**

The paper develops a sampling theory, aiming to decouple the effect of LT on representation
and classifier layers.

Reviewers were concerned with the clarity and novelty of the paper, and pointed out missing
theoretical analysis, experiments, metrics, and baselines. All reviewers found that the paper is
not ready for publication.

No rebuttal was submitted.


**Summary Of Ac-Reviewer Meeting:**

N/A